# Implications of Poly(A) Tail Processing in Repeat Expansion Diseases

**DOI:** 10.3390/cells11040677

**Published:** 2022-02-15

**Authors:** Paweł Joachimiak, Adam Ciesiołka, Grzegorz Figura, Agnieszka Fiszer

**Affiliations:** Department of Medical Biotechnology, Institute of Bioorganic Chemistry Polish Academy of Sciences, 61-704 Poznań, Poland; pjoachimiak@ibch.poznan.pl (P.J.); a.k.ciesiolka@gmail.com (A.C.); grzegorz.figura5@gmail.com (G.F.)

**Keywords:** alternative polyadenylation, repeat expansion diseases, polyglutamine diseases, Huntington’s disease, poly(A) tail

## Abstract

Repeat expansion diseases are a group of more than 40 disorders that affect mainly the nervous and/or muscular system and include myotonic dystrophies, Huntington’s disease, and fragile X syndrome. The mutation-driven expanded repeat tract occurs in specific genes and is composed of tri- to dodeca-nucleotide-long units. Mutant mRNA is a pathogenic factor or important contributor to the disease and has great potential as a therapeutic target. Although repeat expansion diseases are quite well known, there are limited studies concerning polyadenylation events for implicated transcripts that could have profound effects on transcript stability, localization, and translation efficiency. In this review, we briefly present polyadenylation and alternative polyadenylation (APA) mechanisms and discuss their role in the pathogenesis of selected diseases. We also discuss several methods for poly(A) tail measurement (both transcript-specific and transcriptome-wide analyses) and APA site identification—the further development and use of which may contribute to a better understanding of the correlation between APA events and repeat expansion diseases. Finally, we point out some future perspectives on the research into repeat expansion diseases, as well as APA studies.

## 1. Introduction

The main role of mRNAs is to transfer genetic information from DNA into proteins; however, mRNA also functions to provide an additional level of gene expression regulation in cells. Newly generated transcripts-before becoming translation templates-undergo modifications and processing, including splicing and the site selection of the transcript’s 3′ end by the generation of a poly(A) tail [1]. As a result of the latter, a single mature mRNA has a defined 3′-UTR region and a specific poly(A) tail length. Nevertheless, the overall pool of generated mRNAs contains molecules with different 3′ ends. These features differ for any given mRNA-especially in the developmental stages and across various cell types-which affects its stability and overall transcript functioning.

There is a great need for the precise, molecular characterization of transcript variants to understand cell-type-specific processes and their disruption in cases of gene mutations. This includes aspects of poly(A) tail processing, which is especially crucial in neuronal cells, as features of the 3′ end of an mRNA play an important role in regulating the dynamics of RNA metabolism and translation [2].

In this review, we discuss poly(A) tail processing in a context of a group of neurodegenerative and neuromuscular diseases caused by genes mutations consisting of the expansion of a sequence of repeats, e.g., CAG or CUG. A better understanding of a poly(A) tail processing in the context of neuromuscular pathology may provide valuable insight into molecular pathomechanisms triggered by transcript misprocessing and help design RNA-targeted therapies.

We briefly present some general information regarding the processes associated with polyadenylation and the methods used for their analysis. We also discuss previously collected information concerning poly(A) tail processing in mRNAs implicated in repeat expansion diseases, as well as the identified disruptions in this process. This includes the analysis of alternative polyadenylation sites for a group of transcripts containing expanded CAG repeat tracts implicated in polyglutamine (polyQ) diseases.

## 2. Polyadenylation in Animal Cells

### 2.1. Polyadenylation Process

During polyadenylation, a polyadenosine sequence-namely, a poly(A) tail-is added to the 3′ end of a transcript. Together with the removal of introns and the addition of a 5′ cap, polyadenylation constitutes a major step in pre-mRNA maturation [1,3,4]. The polyadenylation process can be divided into two major steps: first, newly transcribed pre-mRNA is cleaved and its 3′ end is generated; then, a specific enzyme-poly(A) polymerase (PAP)-generates the poly(A) tail independently from the template, starting from the cleavage site [5,6,7]. These two steps depend on the interplay between various *cis*- and *trans*-acting factors (Figure 1). *Cis* elements, which are typically located in 3′-UTRs, determine the position of cleavage and polyadenylation site in mRNA [8]. The most crucial *cis*-element is an RNA sequence motif called the polyadenylation signal (PAS). In animals, the canonical PAS is the hexanucleotide sequence AAUAAA [9,10,11]. The cleavage site is located around 10 to 30 nucleotides downstream from the PAS, between the PAS and a core downstream sequence element (DSE)-a U-/GU-rich sequence, which is itself located 14–70 nucleotides downstream of the cleavage site (Figure 1) [12]. The cleavage itself occurs just before an adenosine residue, mostly after cytosine. Another *cis*-element that is involved in the polyadenylation process is a U-rich/UGUA upstream sequence element (USE) located upstream of PAS. All three *cis*-elements are recognized by their respective *trans*-acting factors, which interact with each other through the carboxyl-terminal domain (CTD) of RNA polymerase II (RNAPII) [5]. PAS recruits cleavage and polyadenylation specificity factor (CPSF), DSE is recognized by cleavage stimulation factor (CstF), and USE recruits cleavage factor I_m_ (CFI_m_). Cooperation between those factors leads to cleavage. The subsequent cooperation between nuclear poly(A)-binding protein (PABPN1) and PAP allows for the generation of the poly(A) tail at the cleavage site: PABPN1 acts as a type of ‘ruler’, which is important for the synthesis of an appropriately sized poly(A) tail, while PAP performs a non-templated addition of adenosine residues [13].

### 2.2. Polyadenylation Roles

The poly(A) tail plays an important role in a transcript’s life cycle [2]. For example, it is responsible for mediating the transport of mature mRNA into the cytoplasm through the NXF1-dependent pathway using nuclear-pore complexes embedded in the nuclear envelope [14]. Poly(A) tails are also involved in maintaining the stability of transcripts. They participate in the regulation of mRNA degradation through the process of deadenylation, which is commonly activated in post-transcriptional regulation via the sequence-specific binding of microRNA (miRNA), together with miRNA-induced silencing complex (miRISC), to the 3′-UTR [15,16]. Poly(A) tails are also important for translation regulation [17,18]. It is suggested that they interact functionally and physically with the 5′ cap of mRNA, creating a closed-loop structure that promotes the initiation of translation.

In metazoa, almost all mRNAs undergo polyadenylation. One of the exceptions is replication-dependent histone protein mRNAs, which form a highly conserved stem-loop structure [19,20]. Additionally, according to a recent comprehensive study [21], nearly 50% of long non-coding RNAs (lncRNAs) undergo polyadenylation, and the resulting poly(A) tails are important for their regulation at the cellular level [22].

### 2.3. Length and Composition of Tails

The lengths of poly(A) tails can vary between transcripts. For example, mRNAs of highly expressed genes, such as housekeeping genes, usually possess shorter poly(A) tails, whereas poorly translated transcripts and lncRNAs have longer tails [23,24]. Generally, the length range for specifying a short or long tail is dependent on the population of transcripts analyzed. For example, in a *Caenorhabditis elegans* study, half of the transcripts possessed poly(A) tails in the range of 70 to 94 A residues; therefore, short tails were defined as those with ≤70 A residues, and long ones, as those with >94 A residues [23]. The existence of shorter tails may be also explained by the fact that some genes (e.g., albumin and transferrin) contain poly(A)-limiting element (PLE), which tends to restrict the initial length of poly(A) tails from the pre-mRNA stage [25,26]. Additionally, a cytoplasmic polyadenylation element (CPE) can modulate the length of a poly(A) tail after the export of the transcript from the nucleus [27]. It is located in the 3′-UTR, near PAS, and its most common sequence is UUUUAU [28]. As investigated in neurons and during early development, CPE can influence the length of the poly(A) tail through the binding of proteins to this motif (e.g., CPEB) [29]. CPEB can promote either cytoplasmic polyadenylation or deadenylation. When it is deactivated (dephosphorylated), it recruits poly(A) ribonuclease deadenylase (PARN) to deadenylate and repress the expression of mRNA. On the other hand, when it is phosphorylated, it promotes the expulsion of PARN, thereby leading to polyadenylation by germ-line development factor 2 (GLD2) [26,30]. This cytoplasmic polyadenylation process occurs in mRNAs that already contain a short poly(A) tail and usually activates translation, leading to increased protein expression.

An interesting fact is that poly(A) tails are not limited to possessing A residues: reports indicate that cytosines, guanosines, and uridines can also be incorporated into the poly(A) tail. While the role of cytosines added to the poly(A) tail remains to be elucidated, the guanylation and uridylation of poly(A) tails are quite well understood. Guanylation occurs only for longer poly(A) tails and can slow down deadenylation, delaying transcript decay [31]. Guanylation frequently occurs for long-lived transcripts with a slow turnover, e.g., transcripts encoding secreted proteins [31]. On the other hand, uridylation is usually found in short tails and marks transcripts for decay [32,33]. This process tends to be a key factor in germline development, differentiation, and early embryogenesis, where short-lived transcripts, with relatively fast turnover, predominate [33]. The lengths of poly(A) tails might also depend on circadian rhythms and the cell cycle [34,35,36]. By studying multiple mouse liver mRNAs, researchers demonstrated that rhythmic changes in poly(A) tail lengths were under the control of the circadian clock. Even more importantly, they presented data indicating that rhythmic poly(A) tail lengths are correlated with rhythmic protein expression [34]. With regard to the cell cycle, TAIL-Seq analysis suggests that global RNA decay takes place during the S phase through the accumulation of terminal uridylation. On the other hand, the accumulation of terminal guanylation occurs during the M phase of the cell cycle, leading to the assumption that the majority of the transcriptome is then protected from active deadenylation [35].

### 2.4. Alternative Polyadenylation (APA)

Apart from the canonical PAS-the AAUAAA hexamer-other weaker signals called alternative PASs, may be present in transcripts. Generally, the higher the sequence similarity between an alternative and canonical PAS, the stronger the recognition of the alternative PAS. When this alternative PAS is selected as a signal for the cleavage and polyadenylation event, the process is described as alternative polyadenylation (APA). APA is thought to occur for around 70% of human protein-coding genes and can also affect non-coding RNAs, such as lncRNAs [37,38,39]. The affected transcripts can exhibit various numbers of APA events in a few or multiple APA sites. APA can dramatically modulate the expression of a specific gene and affect the fate of its transcript, including its half-life and cellular localization [40,41]. Depending on the alternative PAS localization, APA can occur either in 3′-UTRs (UTR-APA) or upstream of the last exon: in introns or protein-coding exons (UR-APA). UR-APA can lead to the production of truncated proteins with different functions (protein diversification), or the production of dysfunctional proteins. On the other hand, when APA occurs in the 3′-UTR of a transcript, it leads to the creation of an mRNA of different lengths, which still codes for a full-length protein. In such cases, APA can affect the expression of a gene by, for example, changing the number of miRNA-binding sites in the transcript. As it was shown that the 3′-UTR can regulate protein localization independently from mRNA localization, it can act as a scaffold for various protein complexes which, when recruited to translation sites, can interact with specific domains of newly translated proteins [42]. This, in turn, leads to the translocation of such proteins. An example is a CD47 transcript, whose short 3′-UTR promotes the localization of the protein at the ER, while its longer isoform promotes its translocation to the plasma membrane [37]. The occurrence of APA can be regulated in many ways, one being the ‘strength’ of alternative PASs. The more similar the sequence of an alternative PAS is to that of the canonical PAS, the stronger the alternative PAS will be. Moreover, the localization of a specific PAS within a transcript sequence is also worth mentioning. Typically, PASs localized closer to the start codon (proximal) are considered to be weaker, while PASs localized further from the start codon (distal) are stronger [5]. Core polyadenylation factors, as well as other RNA-binding proteins (RBPs), can also regulate APA. For example, PABPN1 enhances the selection of distal PASs by competing with cleavage and polyadenylation complexes: it recognizes a weak PAS and binds to it, thereby blocking CPSF binding [43,44]. Regarding RBPs, HuR protein favors the selection of a distal PAS by binding to U-rich elements lying close to a proximal PAS [45]. Another crucial group of RBPs involved in APA is muscleblind-like (MBNL) proteins, whose binding sites are present in the close vicinity of many PASs. In myotonic dystrophy (DM), MBNL proteins were shown to either activate or suppress polyadenylation at specific sites [46]. PABPN1, MBNL, and HuR proteins are described in more detail in the following chapters.

## 3. Methods for Determining the Length of Poly(A) Tails and Analyzing APA Sites

Here, we briefly summarize some of the available methods for measuring poly(A) tail lengths using both transcript-specific and transcriptome-wide approaches, as well as methods designed for APA site identification and the currently available APA databases.

### 3.1. Transcript-Specific Analyses of Poly(A) Tails

For transcript-specific analyses, RNase H/Oligo(dT) northern blotting was once the gold-standard technique. Oligo(dT) were hybridized with transcripts, followed by RNase H treatment. Next, poly(A) tail length was assessed through the comparison of transcripts with, or without, hybridized oligo(dT) using northern-blot-based detection [47]. One advantage of this technique was that it avoided introducing a PCR-based bias. However, it had many disadvantages, one being the requirement of large amounts of RNA. This technique is also laborious and works best on highly expressed and shorter mRNAs. These disadvantages encouraged scientists to develop more effective and more precise methods that allow for the transcript-specific measurement of poly(A) tails. Such techniques are usually grouped together and described as poly(A) test (PAT) assays, which include PCR-based rapid amplification of cDNA ends PAT (RACE-PAT), ligation-mediated PAT (LM-PAT) [48], and extension PAT (ePAT) [49]. These techniques, as well as other assays, were reviewed extensively in [26,50]. Currently, the most popular assay is probably ePAT. Here, a 5′-anchored oligo(dT) primer serves as a template for Klenow polymerase, which extends the transcript on the basis of the anchor sequence. This oligo(dT) then serves as a reverse primer for cDNA synthesis. When cDNA is generated, this technique requires the use of a target-specific forward primer for PCR. The PCR products can then be visualized using agarose gel electrophoresis or capillary electrophoresis. Another variation of the ePAT method is the poly(A) tail-length assay, currently available as a commercial kit [51]. This assay uses PAP, which adds guanosine and inosine residues at the 3′ end of every poly(A) tail that exists in the RNA sample. This added G/I tail then becomes a priming site for cDNA synthesis and the following PCR. These PAT methods are fast and easy to perform but require the optimization of the PCR. Otherwise, they may generate PCR-introduced biases and shorter products (i.e., those containing shorter poly(A) tails) may be amplified more efficiently.

### 3.2. Transcriptome-Wide Methods for Measurement of Poly(A) Tails

One method that can determine the lengths of poly(A) tails in both transcript-specific and transcriptome-wide manners is Nanopore technology sequencing [19,52]. Nanopore technology offers a set of products that allow for DNA and RNA sequencing, including for transcripts containing poly(A) tails. The principle of all Nanopore sequencing is that specific adaptors are attached to RNA or DNA, which then allow for such nucleic acids to be pulled through the pore at a constant rate. The length of a poly(A) tail is then estimated based on the correlation between length and the time spent being transferred through the pore. Thanks to numerous approaches, users can decide whether they want to use PCR amplification or bypass this step (to reduce PCR-introduced bias) and sequence the RNA or cDNA with only minimal library preparation, etc. This method is currently one of the best for measuring poly(A) tails and allows for the precise assignment of an analyzed poly(A) tail to a specific transcript. However, it requires large amounts of purified RNA and cannot identify non-A residues within the tail.

Other very popular methods for the transcriptome-wide measurement of poly(A) tail lengths include TAIL-Seq [53] and PAL-Seq [24]. Both methods are very similar in terms of their basic concepts: in the first step, RNAs lacking poly(A) tails (mainly rRNAs) are removed from a pool of sequenced transcripts. Moreover, biotinylated 3′ adaptors are used to allow for the pull-down of bound transcripts on streptavidin beads, which is followed by the addition of 5′ adaptors in both techniques. The main difference between TAIL-Seq and PAL-Seq is their sequencing methodology, which leads to slightly different results. TAIL-Seq allows for the determination of any non-A residues that might be present in the poly(A) tail (i.e., possible guanylation or uridylation). However, this technique is expensive and technically challenging. On the other hand, PAL-Seq permits the capture of tails with only adenosines present at their 3′ ends, but requires less starting RNA material and, due to the enrichment of polyadenylated RNAs at the start of the procedure, is more efficient than the TAIL-Seq approach [19]. PAL-Seq also requires a highly specialized experimental setup and can only be performed on the Illumina platform. In light of these disadvantages, another method that combines the enrichment of polyadenylated RNA from PAL-Seq with the sequencing methodology and algorithm of the TAIL-Seq protocol was developed. This method was named mTAIL-Seq [54] and requires even less starting RNA than the two previously discussed assays. This method also provides higher sensitivity and is less costly than the previous methods; however, it has a limited ability to capture tails shorter than 8 nt and tails that end with non-A nucleotides. Due to its drawbacks, mTAIL-Seq is predominantly used for measuring poly(A) tail lengths.

Recently, a new technique has been developed: full-length poly(A) and mRNA sequencing (FLAM-Seq) [55]. FLAM-Seq relies on 3′ end RNA extension through G/I tailing, template switching, cDNA amplification, and PCR amplification. The transcript sequences are then determined through PacBio sequencing. It is a fast and simple method that allows for the generation of long reads and provides information on poly(A) tail length, sequences of multiple transcripts, and mRNA isoforms. An outstanding advantage of FLAM-Seq is that it allows for the sequencing of poly(A) tails with the detection of non-A residues inside the tail sequences. Moreover, it generates full-length sequences for thousands of mRNAs with a very low error rate [55].

### 3.3. APA-Identification Techniques and Databases

In discussing APA-identification techniques, we can distinguish between experimental methods and the various computational tools available (Table 1). An example of the former is a massive analysis of cDNA ends (MACE-Seq) [56]. This is an improved version of 3′ single-end mRNA-Seq, based on the sequencing of a single molecule of transcripts. Other experimental methods facilitating APA detection include TAIL-Seq, PAL-Seq, and FLAM-Seq, all of which were described in the previous sub-chapter. Regarding computational APA-identification methods, APAtrap allows for the identification and quantification of APA sites derived from various RNA-Seq data [57]. APAtrap also allows for the identification of 3′-UTRs and extended 3′-UTRs, as well as for distinguishing genes that use various APA sites under different conditions. Other computational tools for detecting APA, such as DaPars [58], scDAPA [59], etc., were previously reviewed in detail by others [11,60].

Over recent years, several databases that gather information about possible APA sites in various transcripts have been created. The data collected there were obtained using the previously mentioned RNA-Seq methods and computational tools. One such database is PolyASite 2.0 [61]. It contains information about PASs based on data generated using numerous sequencing methods for human, mouse, and *C. elegans* transcriptomes [61]. Moreover, PolyASite 2.0 provides information about PASs located in 3′-UTRs, introns, and CDSs. Another database is APADB [62], which contains information about human, mouse, and chicken transcripts of both coding and non-coding genes. The data collected there were generated through the NGS-coupled 3′ end sequencing of thousands of samples (e.g., MACE-Seq). APADB also provides information regarding the loss of bioinformatically predicted miRNA-binding sites, which may occur through APA events themselves. Other databases were well-reviewed recently [11,60].

## 4. Repeat Expansion Diseases

Repeat expansion diseases are a group of genetic disorders caused by expansions of polymorphic tracts of nucleotide sequences in specific human genes [72] (Appendix A). There are more than 40 diseases that are classified as repeat expansion disorders, and this number continues to grow [72,73]. Most of them mainly affect the nervous system, but there are also some diseases that impact the muscular system. The most common expanded sequence motifs are trinucleotides, but longer ones, up to dodecanucleotide units, exist [74]. Repeat expansion diseases display several specific features: they are mostly autosomal dominant diseases, but recent reports indicate that mutations of expanded repeats also cause autosomal recessive diseases, e.g., cerebellar ataxia, neuropathy, and vestibular areflexia syndrome (CANVAS) [73]. Typically, patients are heterozygotes possessing wild-type and mutant alleles, with the latter harboring the expanded repeat tract. With regard to the mutant allele, usually the longer the repeat tract, the earlier the onset of symptoms and the more severe the disease course. However, the number of nucleotide repeats needed for disease onset differs among diseases. Expanded repeat tracts can occur in protein-coding sequences of particular genes, as well as in promoter sequences, introns, or in 5′- and 3′-UTRs (Appendix A). Interestingly, many expanded repeat tracts in RNAs are prone to the formation of stable structures like hairpins or G-quadruplexes [75,76]. These structures affect transcript functioning, e.g., through sequestration of proteins [72], whereas many processes remain to be investigated in this context, including APA.

In this chapter, we shortly summarize the tissue-specific processing of poly(A) tails, focusing on the cell types most affected in repeat expansion diseases. Moreover, we discuss selected diseases in the context of APA and poly(A) tail lengths in the implicated mutant transcripts.

### 4.1. APA and Poly(A) Tail Lengths in Neurons and Muscles

An interesting aspect concerning polyadenylation in neurons is the occurrence of neuron-specific proteins that demonstrate a specific impact on polyadenylation. These proteins include the previously mentioned GLD2 [77], mammalian suppressor of tauopathy 2 (MSUT2) [78], zinc finger CCCH domain-containing protein 14 (ZC3H14) [79], and ataxin-2 (ATXN2) [80]. In terms of the present review, the most interesting protein is ATXN2, as the expanded repeat tract in *ATXN2* is responsible for spinocerebellar ataxia type 2 (SCA2) [81], as well as normal ATXN2 is implicated in amyotrophic lateral sclerosis (ALS) [82]. Regarding polyadenylation, ATXN2 inhibits poly(A) nuclease (PAN) activity and causes the occurrence of longer poly(A) tails [80]. It was also suggested that ATXN2 might stabilize its associated mRNAs by suppressing their decay. Additionally, it may contribute to the APA of its associating mRNAs, thereby supporting activity-dependent translation and neural plasticity [80].

Over recent years, it has been shown that, for many genes, the selection of a particular polyadenylation site in a transcript is tissue-specific [83,84,85]. Regarding neurons, studies performed in *Drosophila*, mouse, and human cells demonstrated a neuron-specific enrichment of long 3′-UTRs in various transcripts [41,86,87,88]. An example of such a phenomenon is *Drosophila’s* mei-P26 transcript, whose 3′-UTR is very short in the testis and 18.5 kb long in the central nervous system (CNS) [86]. A tendency for longer 3′-UTR expression in the CNS was suggested to be limited to neurons and is not the case in astrocytes, microglia, or oligodendrocytes [58,63]. An example of a factor that favors the choice of distal PASs is the neuron-specific RBP Elav (embryonic lethal abnormal visual protein), first identified in *Drosophila* [88,89]. Elav binds to its target mRNA at proximal PASs, blocking cleavage and polyadenylation machinery, thereby promoting the use of distal PASs. In *Drosophila* embryos, in the absence of the Elav protein, several genes lack long 3′-UTRs [90]. Human homologs of the Drosophila Elav protein are the Hu-family proteins: HuR, HuB, HuC, and HuD [91].

Due to neuron-specific features such as their structure (a single axon can extend hundreds of centimeters in length) and function (the synaptic plasticity underlying learning and memory processes), a common feature of these cells is local translation [17,30,92]. This phenomenon is required at synapses to, for example, convert short-term memory into long-term memory. Therefore, neurons have had to develop specific mechanisms for the transport and repression of the translation of transcripts (during their transport) throughout the cell, to its distal compartments (e.g., dendrites) [17,92]. Here, the α-subunit of calcium/calmodulin-dependent protein kinase II (αCaMKII) transcript serves as an example (reviewed in [17]). When needed in dendrites, this transcript is polyadenylated in the nucleus and exported to the soma, where it is deadenylated and repressed. It is transported in this state to dendrites and, upon neuronal stimulation, is polyadenylated again and locally translated. This is also an example of the regulation of poly(A) tail length in neurons, which depends on the cell’s needs.

Regarding the muscular system, RNA-sequencing studies performed in *C. elegans* demonstrated the general use of APA sites in muscles. This research revealed that muscle tissue demonstrates a specific polyadenylation pattern [93]. An example of such a tissue-specific form of post-transcriptional regulation led by APA is the *PAX3* (Paired box gene-3) gene, which is a major regulator of muscle stem-cell development and is required for the formation of limb muscles [94] and in response to stress [95,96]. During myogenic differentiation, *PAX3* expression is silenced by miR-206 [97]. The APA-based use of proximal PAS in myogenic progenitors leads to 3′-UTR shortening of the *PAX3* transcript, which results in such mRNA escaping miRNA-induced gene silencing [97,98,99]. In muscles, RBPs also play an important role in the control of PAS selection and have an impact on the general polyadenylation profile. Among such RBPs we specify MBNL proteins, which are discussed in detail in the DM 1 subsection below.

### 4.2. Characteristics and Perturbations in Poly(A) Tail Processing in Repeat Expansion Diseases

#### 4.2.1. Oculopharyngeal Muscular Dystrophy

A repeat expansion disease with crucial APA implications is dominant oculopharyngeal muscular dystrophy (OPMD), which is characterized by progressive eyelid drooping (ptosis), difficulty in swallowing, filamentous intranuclear inclusions in muscle fibers, and proximal limb weakness [100,101]. The gene encoding PABPN1, mentioned in the first chapter, is directly involved in this disease. The relevant mutation involves the expansion of a GCG repeat tract encoding the polyalanine tract, located at the *N*-terminus of the PABPN1. This protein plays an important role in RNA processing, e.g., it is required for efficient mRNA export from the nucleus [102] and controls the length of poly(A) tails [13]. The mutated version of PABPN1 was shown to sequester the WT protein in nuclear inclusions in the muscle fibers of OPMD patients, altering the function of the WT protein [103]. Moreover, it was recently suggested that the PABPN1 may act as an APA suppressor. This hypothesis was supported by the finding that this protein binds near proximal PASs and therefore suppresses the use of these PASs for cleavage and polyadenylation. Additionally, it was demonstrated that either the downregulation or mutation of PABPN1 resulted in the genome-wide use of proximal PASs, which finally cause 3′-UTR shortening to occur in both mouse models of OPMD and in human cells [43,44].

#### 4.2.2. Fragile X-Associated Tremor/Ataxia Syndrome

Fragile X syndrome (FXS), and its related diseases-fragile X-associated immature ovarian insufficiency (FXPOI) and fragile X-associated tremor/ataxia syndrome (FXTAS)-are caused by the expansion of a CGG repeat tract located in the 5′-UTR of *fragile X mental retardation 1* gene (*FMR1*) [104]. In FXPOI and FXTAS, premutation alleles contain 55–200 CGG repeats; however, this mutation does not inactivate the gene [104]. On the other hand, in FXS, the full mutation in *FMR1* of more than 200 CGG repeats leads to transcriptional silencing of the gene [104]. Normally, *FMR1* encodes a protein (FMRP) which is important in brain development as it plays a role in the functioning of neuronal networks and neuronal plasticity [105,106]. In the 3′-UTR of an *FMR1* transcript, three PASs were identified: one canonical and two alternatives [107]. In premutation alleles, the level of transcript isoforms emerging from alternative PASs is decreased, indicating that *FMR1* premutation affects polyadenylation through APA phenomena and thus might impact pathology. Additionally, it was shown that, in this case, APA produces mRNAs with different poly(A) tail lengths: a transcript with expanded CGG tract had a shorter poly(A) tail than its WT counterpart [107].

#### 4.2.3. Amyotrophic Lateral Sclerosis

ALS is a neurodegenerative disease that causes the death of motor neurons that control skeletal muscle contraction. ALS progresses over time, leading to muscle stiffness, weakness, and finally, paralysis and death. This process takes 2–5 years from the initial appearance of symptoms. ALS can have a sporadic or familial character. To date, the pathogenesis of sporadic cases remains unclear, while familial ones are mainly caused by mutations in *C9orf72*, *SOD1*, *FUS*, or *TARDBP* [108,109].

The most common familial form of ALS is caused by the expansion of GGGGCC repeats in the first intron of the *C9orf72* gene. This intron should be degraded after splicing, but because of the mutation, it accumulates in the nucleus and forms RNA foci [110]. Alternatively, it is transported to the cytoplasm where it undergoes RAN translation which, in turn, creates potentially toxic polypeptides [108,111]. RNA-seq was used to determine the APA status of samples from ALS patients, and the results indicated that many defects exist in the overall polyadenylation of transcripts in ALS. Depending on the brain area and type of ALS, distal or proximal PASs are used: (I) in the case of the cerebellum in familial ALS, proximal, rather than distal, PASs are used, while in sporadic ALS, the use of proximal and distal PASs is almost equivalent; (II) in the frontal cortex in familial ALS, proximal and distal PASs have similar rates of use, while in sporadic ALS, distal PASs are used more than in proximal ones [112]. Similar studies demonstrated that 2.7% of genes have statistically significant deregulation of APA in ALS, with a tendency toward the extension of 3′-UTRs. Gene ontology analysis revealed that most of these genes are responsible for neuron-projection development and cytoskeletal intracellular transport [109]. In ALS, the deregulation of polyadenylation affects many genes and potentially seems to be part of the pathogenesis mechanism.

Other genes whose mutations cause ALS, *FUS*, and *TARDBP*, encode two RBPs: FUS and TDP-43 respectively. In *TARDBP*, the mutations can be localized in different regions, but the majority are in disordered low-complexity regions [113,114]. Moreover, in *FUS*, other mutations that cause ALS have been identified, with the most common being a substitution of R521 [114]. The localization of FUS and TDP-43 mutations in their nuclear localization signals (NLS) causes the depletion of these proteins from the nucleus and leads to their accumulation in inclusions in the cytoplasm. Both proteins control the alternative splicing of genes participating in neuronal development which, in the case of mutation, leads to neurodegeneration [115]. Moreover, FUS and TDP-43 are involved in the polyadenylation process. FUS can control the selection of the polyadenylation site depending on the distance from the FUS-binding site [115]. TDP-43′s role is to repress cryptic exons, thereby stabilizing transcripts. TDP-43 suppresses a cryptic PAS of the *stathmin-2* mRNA, which encodes proteins crucial for the regeneration and growth of axons. The re-localization or depletion of TDP-43 causes the recognition of a premature PAS in the *stathmin-2* mRNA and leads to the production of non-functional transcripts [114]. TDP-43 also controls the PAS selection of its own transcript and, in this way, self-regulates its expression. In the case of an excess of TDP-43 in the nucleus, it inhibits the selection of the most proximal PAS (there are three alternative PASs within the TARDBP sequence), leading to the elongation of a transcript. In such cases, the splicing of additional introns occurs, which is followed by the export of the transcript to the cytoplasm and its degradation by nonsense-mediated mRNA decay. As a result, a decrease in the cytoplasmic level of *TARDBP* mRNA is observed. On the other hand, when there is less nuclear TDP-43, the proximal PAS is selected. In this case, the transcript is transported to the cytoplasm and translated. In ALS patients, mutant TDP-43 is depleted from the nucleus, which leads to an increase in the level of TDP-43 in motor neurons by the mechanism described above. This loop may accelerate the progress of the disease through the formation of cytoplasmic inclusions and disturbances in mRNA metabolism [116].

#### 4.2.4. Myotonic Dystrophy Type 1

DM 1 is primarily characterized by muscle dystrophy or myotonia. A mutation responsible for DM1 is the expansion of CUG repeats present in the 3′-UTR of the *DMPK* gene. RNA containing the expanded CUG tract acts through gain of function mechanisms. Mutated transcripts are mainly localized in the nucleus, where they form RNA foci and sequester RBPs such as MBNL1, -2, and -3; HNRNPH1; CUGBP1; and STAU1 [117,118]. This leads to the trans-deregulation of RNA metabolism-especially APA, miRNA processing, and alternative splicing processes [119,120]. MBNL’s implications in alternative splicing and APA are well-documented [121,122,123,124]. The RNA-binding motif of this protein is composed of four zinc-finger domains [125]. Using this motif, MBNL binds to transcripts and act as an activator or repressor of splicing [126]. It was shown that MBNL1 can regulate alternative splicing in the brain, heart, and muscle in post-natal development [46,127]. In DM1, the sequestration of MBNL1 results in the functional inactivation of the protein, leading to disturbances in the alternative splicing of genes controlled by this protein [128,129]. Concerning APA, in MEF cells with *Mbnl1*, *Mbnl2*, and *Mbnl3* depletion, thousands of polyadenylation sites shifts were detected, which suggests that MBNL proteins act globally on alternative translation events [46]. The importance of MBNL1 sequestration in DM1 pathogenesis was demonstrated in skeletal muscle satellite cells [130]. These cells are responsible for muscle regeneration, and, in cases of DM1, this regeneration is impaired. Moreover, autophagy is increased, lowering the proliferation ability of skeletal satellite cells-a process that plays a key role in the initial steps of cell regeneration in damaged muscle. However, the genome modification of *DMPK*, or overexpression of the MBNL1 protein, can ameliorate proliferation defects in skeletal-muscle satellite cells. Indeed, it was revealed that increases in MBNL1 levels in the cytoplasm partially restored appropriate RNA processing [130].

#### 4.2.5. Huntington’s Disease

Huntington’s disease (HD) is a neurodegenerative disorder caused by the expansion of the CAG repeat tract, encoding the polyglutamine (polyQ) tract, in the first exon of the huntingtin gene (*HTT*) [131,132]. The expansion is dominantly inherited, and patients usually develop symptoms in their mid-thirties. Patients harbor two alleles: the wild-type, containing 10 to 30 CAG repeats, and the mutant allele, with over 40 CAG repeats. The mutation of the CAG tract in *HTT* leads to the degeneration of neurons, which emerges in the motor cortex and striatum and is also substantial in other regions of the brain [132,133]. Huntingtin transcripts were reported to have several isoforms produced via APA. Three of these APA events occur in the 3′-UTR of the *HTT* gene, leading to three isoforms that differ only in their 3′-UTR length, with no impact on the coding sequence [134] (Figure 2a, three arrows marked in 3′-UTR). These 3′-UTR isoforms of the *HTT* transcript include a short isoform (PAS: AGUAAA), which is 10.3 kb long, and a long isoform (PAS: AUUAAA), which is 13.7 kb long [134,135]. The third one, 12.5 kb long, is described as an intermediate 3′-UTR isoform (PAS: AAUGAA) and, as well as the previous isoforms, is conserved between mice and humans [134]. This research also suggested that all of these *HTT* isoforms may have different half-lives and localization, as well as RBP- and miRNA-binding sites. Short and long 3′-UTR isoforms are considered to be the most common *HTT* isoforms: the short one is more abundant in actively dividing cells (e.g., B cells and the HEK 293 cell line) and peripheral tissues (muscles), while the long one predominates in non-dividing cells (i.e., terminally differentiated cells), such as in brain tissue, but also prevails in breast and ovary tissues [134,135]. Research using SH-SY5Y cells and focusing on poly(A) tail length in *HTT* transcripts revealed that each of these three *HTT* mRNA isoforms had different lengths of poly(A) tails. The short 3′-UTR isoform possesses around 60 adenosine residues in its poly(A) tail, while the intermediate and long 3′-UTR isoforms have around 5 and 10 A residues, respectively [134]. These results suggested that *HTT* transcripts have rather short poly(A) tail lengths, at least in this specific cell line. It was also shown that these isoforms arise from both alleles (normal and mutant) in HD, meaning that CAG expansion probably does not have an impact on PAS selection in the 3′-UTR of *HTT* [134]. However, another study concluded that the short 3′-UTR isoform might be translated more efficiently than the longer one [136].

In HD, an additional isoform of the transcript (*HTTexon1)* is produced predominantly from mutant *HTT* due to APA site selection in intron 1 [137] (Figure 2a, red arrow in the first intron of *HTT*). This APA event results in an isoform of 7.9 kb, which is translated into the highly truncated huntingtin protein, with a high tendency to form aggregates. This mRNA isoform has been detected in HD patients but not in healthy individuals [137]. Another study revealed higher levels of this isoform in the HD patient hippocampus and motor cortex compared with controls [138]. Those results strongly suggest that the production of *HTT* mRNA containing exon 1 and part of intron 1 is associated with HD. One hypothesis that might explain this generation of such an mRNA isoform is related to the serine/arginine-rich splicing factor 6 (SRSF6). Bioinformatic analysis predicted that SRSF6 binds to CAG and CAGCAAA repeats, resulting in the incomplete splicing of exon 1 to exon 2 (intron 1 retention) and simultaneously leading to the exposure of cryptic poly(A) sites within intron 1 [139]. However, a subsequent in vivo study performed by the same group demonstrated that SRSF6 is not needed for the aberrant splicing of *HTT* [140], meaning that this protein and its sequestration by CAG repeats is not responsible for a mechanism leading to the formation of *HTTexon1* mRNA, which warrants further investigation.

### 4.3. Potential Perturbations in Poly(A) Tail Processing in Other PolyQ Diseases

HD is one of nine polyQ diseases that are all caused by CAG repeat expansion located in the ORF regions of specific genes [141]. Due to already-known perturbations in poly(A) tail processing in *HTT*, as described in the previous subchapter, we more closely examined other transcripts directly involved in polyQ diseases. Apart from *HTT*, we searched the PolyASite 2.0 database [61] for information concerning alternative polyadenylation sites in eight other mRNAs: *AR*, *ATN1*, *ATXN1*, *ATXN2*, *ATXN3*, *CACNA1A*, *ATXN7*, and *TBP*, whose genes contain mutations responsible for spinal and bulbar muscular atrophy (SBMA), dentatorubral-pallidoluysian atrophy (DRPLA), and spinocerebellar ataxia types 1, 2, 3, 6, 7, and 17 (SCA1, SCA2, SCA3, SCA6, SCA7, and SCA17, respectively). For all the mRNAs, 3′ end sequencing data were available and included a set of sites for alternative polyadenylation (Figure 2b–i). Apart from 3′-UTRs, identified PASs were located in protein-coding exons and introns, but the use of these sites was often quantified as very low. Nevertheless, the identification of these sites suggests the possibility of premature transcription termination (PTT) [142] for these transcripts which may specifically occur under certain conditions. Selected sites are presented in Figure 2, where we include a presentation of PASs with relatively high TPM values (≥0.1) within all the gene regions, as well as all the PASs that fulfilled at least one of the following criteria: (I) presence in the same exon as a CAG repeat tract, (II) presence in an exonic sequence at a distance of less than 500 nt from a CAG repeat tract, or (III) presence in introns directly adjacent to an exon containing CAG repeat tract. The purpose of these criteria was to verify the presence of PASs that could potentially be of importance in cases of repeat-tract mutations. Indeed, numerous PASs were identified according to the listed criteria for most analyzed mRNAs. Abnormal interactions of proteins with expanded CAG repeats in RNA, or the abnormal secondary structure of this mutated region [143], can potentially lead to disruptions in splicing and/or PAS recognition. In several of the investigated transcripts, PASs were present in neighboring introns, which could lead to the production of a shorter transcript and smaller protein if a splicing aberration—similar to that reported for *HTT*—occurs. These effects may be crucial for pathogenesis, especially for mRNAs where the CAG repeat tract is located closer to the 5′ end of a transcript, such as in *HTT*. For mRNAs including *AR*, *ATXN2*, and *TBP*, if the resulting transcript would still contain the mutant CAG tract, it would be translated into a very short aberrant protein with expanded polyQ tracts that could be very prone to aggregation. These events are speculative at present, but, as was shown in HD, may be important contributors in cascades of pathogenic events.

In a set of analyzed mRNAs, there exists variation in the number and distribution of PASs in the 3′-UTRs (Figure 2). Most of the transcripts have more than one PAS identified in the 3′-UTR. As previously mentioned, one of the consequences of the selection of proximal or distal PASs, especially in the 3′-UTR, is the presence or absence of miRNA-binding sites. We included additional analysis of the data gathered in APAdb [62] to list miRNAs that lose their binding sites in the case of proximal or intermediate PAS selection in *HTT* (Figure 3a) and *ATXN3* (Figure 3b) mRNAs. Numerous miRNAs are identified as potentially regulating these transcripts and potential proximal PAS selection will lead to a lack of these expression-regulating sites.

## 5. Conclusions and Perspectives

3′-UTRs, where polyadenylation events usually occur, had their lengths generally increased during evolution and therefore are considered as one of the elements of increased molecular, and as a result also morphological, complexity [145]. APA adds another level for the regulatory role of 3′-UTRs, as transcripts that substantially differ in 3′-UTR length can be generated. This results in the presence, or absence, of specific structural elements, RNA- and protein-binding regions, etc., and thus it affects the metabolism of a given transcript [146]. 

APA has various biological functions, both physiological and pathological. The most recent data suggest that APA can be a molecular tool for regulating mRNA and protein levels during cellular responses, e.g., differentiation [38]. On the other hand, as APA can affect the expression of various genes and can produce incorrect proteins, it is also associated with many diseases, whether oncological, immunological, endocrine, or neurological.

### 5.1. Methodological Aspects

As poly(A) tail processing is involved in many physiological and pathological mechanisms, there is still a great need for the development of various tools and methods that will allow for more insightful studies of APA sites and tail lengths. It has now become possible to precisely analyze polyadenylation at the single-cell level [41,147,148], but data gathered from multiple cell types and at different developmental stages is needed to advance our understanding of events connected to polyadenylation. Moreover, this process is clearly associated with other molecular mechanisms that occur for specific transcripts, and a further consolidation of the available data is required to provide a comprehensive view of transcript functioning.

### 5.2. Implications for Pathogenesis of Repeat Expansion Diseases

As the majority of transcripts undergo the selection of APA sites, it remains to be elucidated how eventual mis-selection events are connected with diseases. In mutant transcripts, repeat tract expansion can affect PAS selection, as was described in HD. Repeat expansion diseases mainly affect specific cell types, e.g., a specific vulnerability of neuronal cells to degeneration is observed [149]. Gathering more and more data from disease tissues and cells will enable the identification of potential disruptions which could occur in specific cell types in cases of repeat tract mutation. Additionally, as widespread APA deregulation has been already observed in a range of neurological disorders [109], it would be of advantage to establish what are the specifically deregulated APA events also in repeat expansion diseases. This knowledge could be used for molecular diagnostics, as it has been recently suggested that APA-signatures in cancer appear to outperform existing biomarkers [58,150,151]. 

### 5.3. Implications into Therapy of Repeat Expansion Diseases

All repeat expansion diseases are as yet incurable, and often, especially for diseases where the expanded repeat tract is located in a coding sequence, the most straightforward therapeutic approach would be to eliminate the expression of the mutant gene. Many challenges remain to be faced in the development of these strategies. A crucial aim is to advance our knowledge regarding the therapeutic target features that could allow for the design of the most efficient and safest molecules for therapy. Mutant transcripts are currently the most promising targets, as various tools can be designed to bind to them in order to inhibit mutant gene expression. In regards to the therapeutic aspects, it remains to be determined whether specific 3′ end transcript variants are present in cases of repeat tract mutation, and, if so, how they could be specifically targeted. Specific therapeutic interventions based on modulation of biogenesis or direct elimination of faulty APA isoforms are being currently extensively developed for other diseases such as cancer (as reviewed in [3]). Currently, the most desirable therapeutic tools for the majority of repeat expansion diseases are allele-selective ones, which preferentially target the expression of the mutant allele. Various types of oligonucleotides-including short interfering RNAs (siRNAs) and antisense oligonucleotides (ASOs)-have been tested in animal models, and clinical testing is ongoing [152]. Interestingly, some of the mutant mRNA-targeting oligonucleotides may cause deadenylation. Our recent study, focusing on the mechanisms leading to the allele-selective silencing of *HTT*, demonstrated that a specific self-duplexing siRNA (sd-siRNA) [153], which is described as miRNA-like siRNA, caused the rapid deadenylation of the mutant transcript and translational inhibition [154]. It remains to be investigated how deadenylation is activated in this therapeutic strategy using an endogene of *HTT*.

It seems we are greatly advancing our understanding of how important the poly(A) tail-connected events are in the molecular biology puzzles. In repeat expansion diseases, we already know some relationships of APA to selected diseases, but it looks like a lot is ahead of us to unravel. 

## Figures and Tables

**Figure 1 cells-11-00677-f001:**
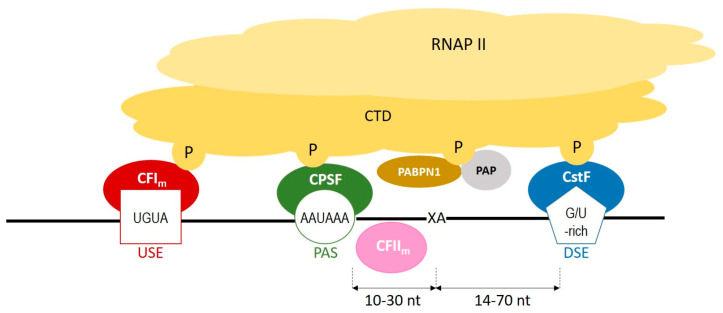
A scheme representing the cooperation between *cis*-elements and *trans*-factors, which are involved in a polyadenylation process. CFIm–cleavage factor Im; CFIIm–cleavage factor IIm; CPSF–cleavage and polyadenylation specificity factor; CstF–cleavage stimulation factor; CTD–carboxyterminal domain of RNA polymerase II; DSE–downstream element; P–phosphorylated serines of CTD; PABPN1–nuclear poly(A) binding protein; PAP–poly(A) polymerase; PAS–poly(A) signal; RNAPII–RNA polymerase II; USE–upstream element; XA–cleavage site.

**Figure 2 cells-11-00677-f002:**
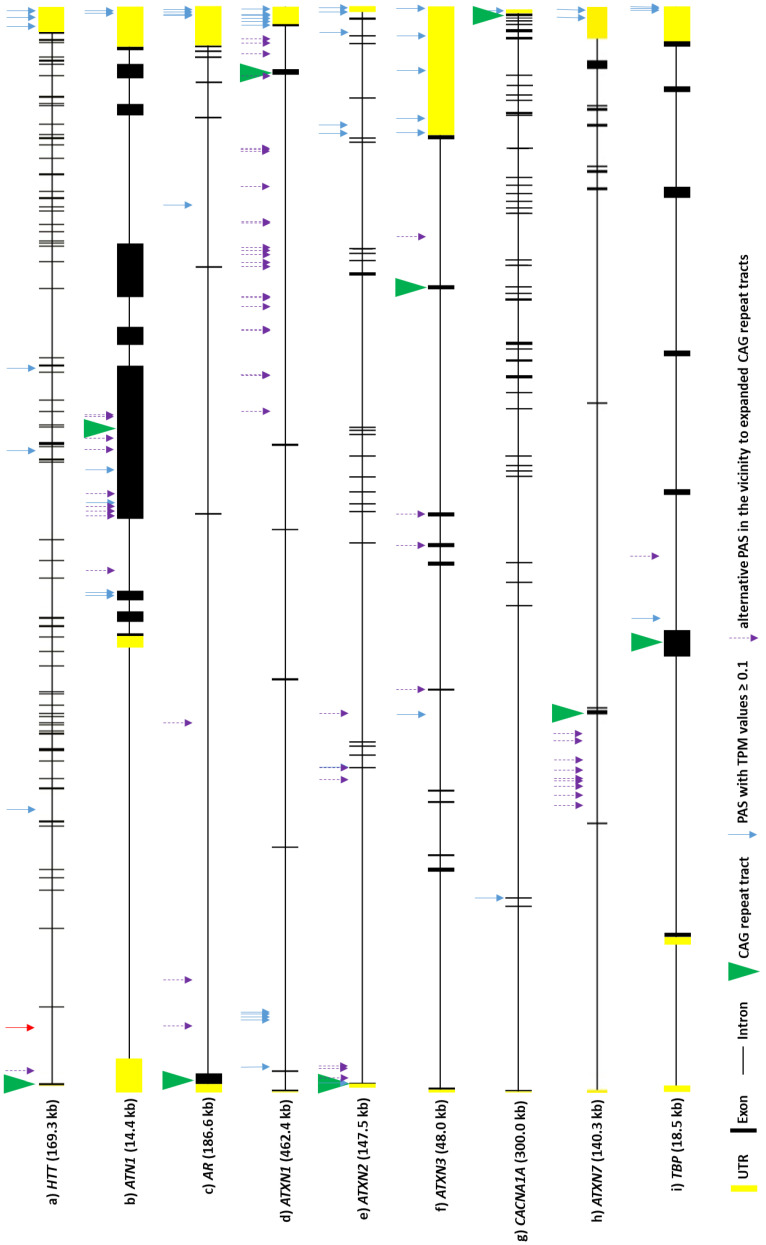
A scheme representing genes associated with polyQ diseases, based on Ensembl data [144], with marked alternative polyadenylation sites derived from PolyASite 2.0 database. The red arrow represents PAS identified in intron 1 of mutant *HTT* gene. See text for more details.

**Figure 3 cells-11-00677-f003:**
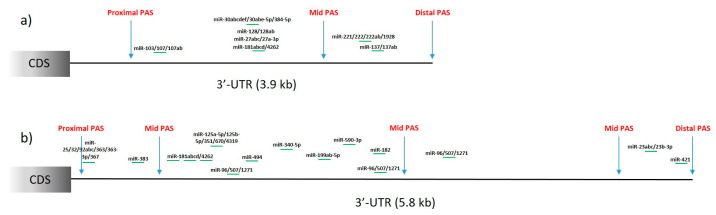
A schematic representation of miRNA-binding sites that are present in 3′-UTRs of *HTT* (**a**) and *ATXN3* (**b**) mRNAs with available PASs marked. Selection of PAS other than distal will lead to a lack of specific miRNA-binding sites.

**Table 1 cells-11-00677-t001:** Summary of experimental methods and computational tools allowing for APA sites detection together with APA databases (accessed on 12 February 2022).

Name	Year of Publication	Website	References
**Computational tools for APA detection**			
DaPars	2014	https://github.com/ZhengXia/dapars	[58]
Change point	2014	http://utr.sourceforge.net	[63]
Roar	2016	https://github.com/vodkatad/roar	[64]
APAtrap	2018	https://apatrap.sourceforge.io	[57]
QAPA	2018	https://www.github.com/morrislab/qapa	[65]
TAPAS	2018	https://github.com/arefeen/TAPAS	[66]
KAPAC	2018	https://github.com/zavolanlab/PAQR_KAPAC.git	[67]
scDAPA	2019	https://scdapa.sourceforge.io	[59]
**APA databases**			
PolyA-Seq Atlas	2012	http://genome.ucsc.edu/	[10]
APADB	2014	http://tools.genxpro.net:9000/apadb/	[62]
APASdb	2015	http://genome.bucm.edu.cn/utr/	[68]
PolyA_DB3	2018	https://exon.apps.wistar.org/PolyA_DB/v3/	[69]
APAatlas	2020	https://hanlab.uth.edu/apa/	[70]
PolyAsite 2.0	2020	https://polyasite.unibas.ch	[61]
Animal-APAdb	2021	http://gong_lab.hzau.edu.cn/Animal-APAdb/	[71]

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
