# Peer review of "Implications of Poly(A) Tail Processing in Repeat Expansion Diseases"

_cells, 2022, doi:10.3390/cells11040677_

Round 1

Reviewer 1 Report

The authors brief review the process of polyadenylation and alternative polyadenylation, summarizes methods for investigating the poly(A) and APA databases, and discuss the role of polyadenylation and alternative polyadenylation in a number of repeat expansion diseases. This manuscript will be of interest to the repeat expansion disease field.

I have a few minor comments:

  1. First sentence of abstract,”…affect the nervous and/or muscular system”, it would be better to add a word “mainly” in the sentence to be rigorous.
  2. Line 294, “With regard to the mutant allele, the longer the repeat tract, the earlier the onset of symptoms and the more severe the disease course.” It is true for some repeat expansion diseases at certain repeat range. The sentence needs to be modified to make it rigorous and accurate.
  3. Line 595,”…the most straightforward therapeutic approach would be to eliminate the expression of the mutant gene.” The sentence needs to be modified. It is not true for all repeat expansion diseases. For example, the fragile x syndrome is caused by eliminate the expression of FMRP protein.
  4. Table S1, add the full name of disease will be helpful.

Reviewer 2 Report

Authors present a timely review on polyadenylation processing in repeat associated disorders. It is a well written manuscript except that the introductory part (which has been reviewed several times elsewhere) is quite long and disease part lacks a detailed mechanistic discussion- this does not make it a poor review though. 

I would suggest to bring up a discussion on any possible association of RNA structural elements in 3'UTR (such as G-quadruplexes and/or hairpins), alternate PAS, and RNA decay. 

Figure 1- color choice is not perfect.
